# Myosin and Other Energy-Transducing ATPases: Structural Dynamics Studied by Electron Paramagnetic Resonance

**DOI:** 10.3390/ijms21020672

**Published:** 2020-01-20

**Authors:** Toshiaki Arata

**Affiliations:** Department of Biology, Graduate School of Science, Osaka City University, Osaka 558-8585, Japan; arata@sci.osaka-cu.ac.jp; Tel.: +81-6-6605-3158

**Keywords:** myosin, kinesin, troponin, tropomyosin, P-type ATPase, KaiC, spin labeling, protein structural dynamics, EPR spectroscopy, energy transduction

## Abstract

The objective of this article was to document the energy-transducing and regulatory interactions in supramolecular complexes such as motor, pump, and clock ATPases. The dynamics and structural features were characterized by motion and distance measurements using spin-labeling electron paramagnetic resonance (EPR) spectroscopy. In particular, we focused on myosin ATPase with actin–troponin–tropomyosin, neural kinesin ATPase with microtubule, P-type ion-motive ATPase, and cyanobacterial clock ATPase. Finally, we have described the relationships or common principles among the molecular mechanisms of various energy-transducing systems and how the large-scale thermal structural transition of flexible elements from one state to the other precedes the subsequent irreversible chemical reactions.

## 1. Introduction

There are many examples in which a protein supramolecule, usually an enzyme (ATPase) or enzyme complex, can exist in distinct states, such that this molecule undergoes continuous cycling among these states at steady state. Ligands, a substrate (ATP), products (ADP, phosphate), and partner proteins bind to enzymes and form chemical intermediate states during hydrolysis. To elucidate how the structural dynamics promote energy conversion, we analyzed the structure of these states using electron paramagnetic resonance (EPR) and spin-labeling. Our results demonstrated that all systems have flexible elements or interconvertible structural states that play a central role in various kinds of motile functions, e.g., active transport, clock, etc. An analysis of this type provides a foundation for understanding the general principles involved in many bioenergetic transformation problems.

## 2. Site-Directed Spin-Labeling EPR Spectroscopy: Advantages and Limitations

A prerequisite for the utilization of EPR spectroscopy is the introduction of paramagnetic centers. For this purpose, several paramagnetic species can be chosen, e.g., radical centers, metal ions, Mn^2+^, Cu^2+^, or Gd^2+^. Due to its specificity to cysteine residues, chemical stability, and only small structural perturbations, the nitroxide spin label 4-maleimido-2,2,6,6-tetramethyl-1-piperidinyloxy (MSL), *N*-(1-oxyl-2,2,6,6-tetramethyl-4-piperdinyl) iodoacetamide (IASL), or (1-oxyl-2,2,5,5-tetramethylpyrrolidin-3-yl) methylmethanethiosulfonate (MTSL) is the most commonly used EPR spin probe in proteins [1,2]. Usually, one or two amino acid is genetically mutated to cysteine [3]. Additional functional inspections are required to confirm the functionality of the mutated, spin-labeled protein for all spin-labeling approaches.

In general, continuous wave (CW)-EPR measurements performed at different temperatures produce diverse information about the spin-label dynamics: the interaction with its local environment. The spin label’s motion is characterized by its rotational correlation time, which depends on the spin label’s intrinsic flexibility and its length. The protein itself tumbles in solution, which determines the overall rotational correlation time dominantly when the spin label is fixed on the protein surface. In particular, the orientational order or disorder of the spin label fixed on the protein can be also determined at a high angular resolution [1,4]. The measured effective rotational correlation time of a spin label bound to a protein is usually a result of the spin label’s interaction with the primary, secondary, tertiary, and eventually quaternary structure of the protein. The term “mobility” can be generally interpreted to describe the interaction of the nitroxide side chain with its local environment. This relatively simple method, when used in conjunction with cysteine scanning and nitroxide movement along peptides of the protein surface, differentiates between the surfaces of proteins with or without a partner protein or domain [1]. This identifies where the binding sites and areas of the protein–protein complex are, even when they are interacting weakly. X-ray crystallography and cryo-electron microscopy are typically used to provide information on buried and fixed side-chains and backbone peptide chains of proteins at atomic resolution for a broad molecular weight range [5,6,7]. On the other hand, site-directed spin-labeling (SDSL) EPR spectroscopy provides side-chain information for the protein surface, which is not obtainable with the former techniques.

Forster fluorescence energy transfer spectroscopy (FRET) and EPR dipolar spectroscopy provide closely related information on inter-label distances in proteins. Pulsed double electron resonance (DEER/PELDOR) and double coherence (DQC) EPR [8,9,10] and FRET detect the same distance range (2–10 nm). The main advantage of FRET is the fact that it can be applied to batch (nanomolar concentrations) and single molecule samples at physiological temperature, while EPR provides distance information on frozen molecular ensembles at micromolar concentrations. However, the smaller size of nitroxide labels in comparison with many fluorophores presents the potential for less perturbation and may allow label placement closer to functionally sensitive regions. Distance determination by EPR method is suitable over a broad range without changing label type, while FRET uses different donor and acceptor labels. In FRET, the orientation between the donor and acceptor molecules must be known, and this is difficult to know. EPR distance measurement does not have this problem with the relative transition dipoles, because the localized magnetic transition dipoles are oriented by the magnetic field, not by the molecular structure. Furthermore, it is also possible to use non-selective pulses, minimizing effects of label orientation by detection of few traces at different magnetic fields.

The fluorescent properties of fluorophores are often affected by their environment, which again complicates exact distance determination. CW-EPR can detect a shorter distance range, 0.8–2 nm, by dipolar interaction, and a further <0.8 nm by exchange interaction. This method is very useful in elucidating the arrangement of α-helices within protein domains by measuring the distance between the side chains of two neighboring α-helices. Cysteine scanning also allows for secondary structure determination. The method, based on the spin–spin interactions between nitroxides attached to the *i, i + 3,* and *i, i + 4* residues, identifies α-helix, 3_10_-helix, and unstructured regions [1,11].

Spin-labeling EPR spectroscopy is very useful and unique for detecting the side chain mobility and the inter-side-chain distance of spin-labeled proteins. However, the information on the backbone structure of the protein is not derived from side-chain information because the spin labels are bound to the side chain and localized even further away, at 0.9–1.5 nm from the backbone of the protein. Both backbone and side chain information are needed to understand the functional movement of a working protein complex. Therefore, validating structural data or modeling protein conformational changes requires simulation of the conformations adopted by the spin probe with respect to the protein backbone, if we compare the EPR distance and mobility with the X-ray crystal or cryo-EM structure. Extensive computational studies are required to fit the distance and mobility data with many unknown parameters in order to comprehensively understand the conformational changes, because backbone data are usually not available for protein complexes during active conformational change. Bifunctional spin labels with two linker groups facilitate attachment to two sites on a protein, reducing probe mobility with respect to the protein [12,13]. The coordination binding of paramagnetic metals also overcomes the above problem [14]. However, we generally did not take such approaches and assumed that the side chain movements were active conformational changes. When the conformational changes modulated tertiary interactions of side chains arising from backbone movement or rotation, they were detected by changes in spin-label mobility: a striking pattern of changes in mobility and overall line shape was observed. The distance change of nitroxide–nitroxide interactions (<1.5 nm) was ideal for mapping nearest neighbor secondary structures and their relative movement. The combined use of mobility and distance changes provided a description of the conformational change at the level of backbone and side chain movement, although precise structural changes could not be determined.

## 3. Myosin ATPase

Muscle contraction is driven by myosin attached to an actin filament. Skeletal muscle myosin has two heads. ATP is hydrolyzed via the following reaction [15]: AM + ATP → (A)M.ATP → (A)M.ADP.P → AM.ADP + Pi → AM + ADP. A number of structural studies have focused on how myosin heads generate a shape change. Fluorescence depolarization [16], electrobirefringence [17], and small-angle X-ray scattering [18] have shown that myosin heads have gyration radii smaller during ATP hydrolysis than in the apo state. This is because of the bending motion of the light-chain-binding region of the myosin heads as a lever arm. Crystallographic studies using phosphate transition analogs have finally shown these at atomic resolution [19]. More recently, Arata et al. (unpublished [20,21]), in collaboration with the K. Wakabayashi group, proposed oppositely bent structures as new intermediate states of ATP hydrolysis. The structures of myosin heads as shown by small-angle X-ray scattering during ATP hydrolysis at low temperatures and in the SH1–SH2 crosslinked state were similar to the crystal structure of the scallop ADP state [19] and the electron microscopic structure of the SH1–SH2 crosslinked state [22] where the SH1–SH2 helix is unwound. The other previously resolved intermediate states of crystal structures showed no indication of SH1–SH2 unwinding. Because it is well known that during hydrolysis, the SH1–SH2 helix is unwound [19], the oppositely bent structure must be a new structure of an intermediate state.

Many studies using probes have attempted to find structural changes in the lever arms of myosin heads bound to actin filaments in skinned or glycerinated fibers. Studies using probes at Cys707 or catalytic sites (Figure 1) have failed to detect two large-scale structurally different and well-defined states during ATP hydrolysis [23,24]. Thomas and colleagues [25] and Arata [26,27] found a weakly actin-bound state with a variety of head orientations. The actin–myosin interface structure of the weakly bound state was studied using monomeric actin to explore whether one or two actins were bound to the myosin head [28,29,30,31]. Arata (1991) [4,32,33] and Irving et al. [34,35] first tried to place a spectroscopic probe at the regulatory light chain and exchange it into muscle fibers. Electron paramagnetic resonance (EPR) spectra from a maleimide nitroxide spin label at the regulatory light chain (RLC) in muscle fibers parallel to the magnetic field showed that myosin head S1, with a spin label at Cys 154 RLC diffused inside, bound to the thin filament and had a well-defined single angle. On the other hand, the two-headed H-meromyosin (HMM) had two angles, suggesting that the neck regions are flexible relative to the catalytic domain. This view was supported by single-molecule fluorescence polarization from the RLC of HMM [36]. The spin label at the RLC of intrinsic myosin heads appeared considerable disordered in muscle fibers in the rigor (−ATP) state, with the corresponding spectra similar to that of random orientation (Figure 2A) (Arata, unpublished [37,38,39]). Similar results were obtained by other laboratories [40,41,42,43,44,45]. However, the difference spectra between rigor and relaxed (+ATP, −Ca^2+^) spectra demonstrated two kinds of well-ordered spectral components. The observed spectra were then decomposed into these two components, with the distributions having the full width of ~40°). One of the two was 1.2-fold more populated than the other in rigor state. In the relaxed state, the two angular populations were nearly even. During isometric contraction, myosin heads oriented again at the two angles of RLC domains, as in the relaxed state. The force was generated by one of two angles (30–50% of heads), which underwent the transition between these two angles upon being bound to actin. Therefore, the myosin molecule shifted the same “gear” (RLC rotation) from relaxed to active state. The two RLC domains of the myosin molecule underwent thermal fluctuation between these two angles in both relaxed and active state. Upon activation, the RLC domain of the actin-bound head underwent a force-producing transition from one angle to the other, while the other head with the second angle dissociated.

The central goal in muscle energy transduction research has been to detect the force-generating structural changes in the actin–myosin complex. Conformational changes within the myosin head during steady-state ATP hydrolysis and in the presence of ATP and phosphate analogs have been detected by a number of methods, as described in the previous paragraph. The conformational states of the myosin head must be correlated with the specific biochemical intermediates and ultimately with the force-generating transition. It is currently proposed that the bending motion of the light-chain-binding region of the myosin heads occurs from the M.ADP.P to M.ADP, state and that M.ADP is a force-generating state that is produced after release of phosphate from MADP.P [46]. However, little is known about how the force generation of myosin and actin is directly coupled to the chemical reactions that drive conformational changes. The first suggestion was provided by Arata and Shimizu (1981) [47], and later by Ostap et al. [48]. They measured EPR spectra from isometrically contracting muscle fibers where myosin was labeled at Cys707 with an iodoacetamide nitroxide spin label (IASL), and found that the force-generating state had a conformation, detected by internal spin-label mobility, that was different from that of the predominant actomyosin (A)M.ADP.P state in solution [49,50] and was rather similar to the AM state (i.e., the mobility of actomyosin without ATP). However, Nagano and Yanagida (1985) [51] measured the fluorescence from the nucleotide bound to myosin in contracting muscle fibers and showed that the force-generating state has a conformation similar to the AM.ADP.P state in solution. These reports suggested that the catalytic site and Cys707 site behaved differently from one another. In other words, structural changes in the converter domain preceded the chemical reaction at the catalytic site. Recently, Thomas and colleagues [52] measured the time-resolved fluorescence energy transfer (FRET) between the relay-helix- or catalytic-site-RLC using the stopped-flow method, and concluded that the structural change preceded phosphate release. It is likely that thermal fluctuation drives the structural change of (A)M.ADP.P (non-force) to (A)M.ADP.P (force), and the latter is dominant under isometric condition. Thermal fluctuation involves the rotation of the RLC domain of the myosin head bound to a site of actin, as described in the previous paragraph. When muscle shortens under low load, the latter accelerates subsequent irreversible phosphate and ADP release, which prevents a backward structural change, and the former (A)M.ADP.P (non-force), which is a weakly actin-bound state, becomes dominant. Therefore, the myosin head behaves as a strain sensor. This mechanism resembles Huxley’s 1957 model [53], where the motor domain undergoes thermal fluctuation and can only bind to its site on the filament when the elastic element is strained. Under no load, thermal fluctuation of a weakly actin-bound (A)M.ADP.P (non-force) may be uncoupled with ATP hydrolysis because the chemical reaction (ADP and Pi release) is slow compared with shortening rate of myofibrils [54,55]. The ATPase measurement showed that the number of ATPs hydrolyzed (/s) by one thick filament of one sarcomere (Z-line-digested myofibril) was much lower than the shortening rate (nm/s) divided by step size (<10 nm). In the non-force state, the RLC domain thermally fluctuates and myosin heads detach and attach rapidly and move forward and backward for long distances. When the myosin head moves forward over a long distance, the RLC domain is assumed to tilt forward to s well-defined, rigor-like (stereospecific) actomyosin-binding orientation (in force state) in a moment. At this moment, the phosphate release occurs and prevents backward movement by strong actin–myosin binding. When the myosin head moves backward, the rigor-like binding orientation is assumed to be attained with difficulty. This hypothesis was recently supported by single-molecule observation of an artificial thick filament [56].

## 4. Regulation of Myosin ATPase by the Actin–Troponin–Tropomyosin Filament

Skeletal and cardiac muscle thin filaments consist of actin, tropomyosin I, and troponin (Tn). Conformational changes in Tn, induced by the binding of Ca^2+^ and in association with Tm, act as a switch that regulates muscle contraction [57,58]. Tn consists of three subunits, troponins C, I, and T. The conformational changes of Tn upon binding of Ca^2+^ have been studied by crystallography, nuclear magnetic resonance (NMR) and FRET, fluorescence polarization, and X-ray/neutron scattering [59,60,61,62,63,64,65,66,67]. We used SDSL-EPR for the structural changes of Tn. The distances between the two spin labels on genetically substituted cysteines of Tn were determined by dipolar EPR, CW-EPR, and pulsed EPR spectroscopies (PELDOR/DEER and DQC). All of the distances determined by these EPR spectroscopies were consistent with those assessed by the other methods, X-ray crystallography and NMR. Aihara et al. [68] measured the distance between the TnC N-lobe and the TnI switch region in a reconstituted ternary TnC–I–T complex (Figure 3). Large distance changes were observed by PELDOR/DEER, concomitant with TnI mobility changes [69]. In the −Ca^2+^ state, the distances displayed broad distributions, showing that the switch region of TnI was physically released from the N-lobe of TnC and consequently fluctuated over a variety of distances on a large scale (2–8 nm). In the +Ca^2+^ state, the TnC–TnI distance had a narrow distribution, showing that the switch region bound to the N-lobe of TnC. These results support the concept that the switch region of TnI, as a molecular switch, binds to the exposed hydrophobic patch of TnC and traps the inhibitory region of TnI away from actin in the Ca^2+^ activation of muscle. However, the switch region of TnI did not behave like a simple two-state switch, because it fluctuated thermally in the −Ca^2+^ state. The activated and inactivated positions coexisted in equilibrium in the −Ca^2+^ state, while there was a single activated position in the +Ca^2+^ state. It is likely that there is a thermal fluctuation of the switch region of TnI between the inactivated and activated states (even in the −Ca^2+^ state), and that the latter state is stabilized by the binding of Ca^2+^ to TnC.

Cardiac Tn (cTnC) has a single functional Ca^2+^-binding site instead of the two Ca^2+^ functional binding sites of skeletal TnC. Ueki et al. [70] found that the interspin distance of the N-lobe of cTnC alone, determined by CW-EPR, showed a small change of ~0.15 nm, suggesting the partial opening and exposure of hydrophobic patches upon Ca^2+^ binding. Interestingly, the N-lobe opened fully with a distance change of ~0.4 nm in the cTnC–cTnI complex [70]. Nakamura et al. [71] measured the interspin distance of the TnC N-lobe in the range of 2–8 nm by PELDOR/DEER in the TnC–TnI complex and muscle fibers into which labeled TnC was exchanged, demonstrating a small change (~0.5 nm) and the opening and exposure of hydrophobic patches upon Ca^2+^ binding. Abe et al. [72] also determined the distance between the cTnC N-lobe and the C-lobe by DQC in the reconstituted cTnC–cTnI complex, demonstrating an interspin distance of 3–5 nm, with at least two distance distributions separated by 0.7–1.5 nm even in the −Ca^2+^ state, of which the shorter distribution populated 20%. Upon Ca^2+^ addition, the shorter distance distribution increased from 20% to 40–50%. It is likely that the regulatory region of cTnI binds to the hydrophobic patch of the cTnC N-lobe in the same manner as the skeletal TnC–TnI complex, and that cTnC then moves its N-lobe down toward the C-lobe. The inactivated (longer distance) and activated (shorter distance) positions coexist even in the -Ca^2+^ state, as in the case of skeletal TnC described above. However, upon Ca^2+^ addition, the fraction in the activated state was not at unity and remained low, at 40–50%, unlike skeletal TnC. Similarly, Potluri et al. [73] measured the distance between the cTnC N lobe and the cTnI switch region, which was smaller relative to the skeletal distances determined by Aihara et al. [68] and described in the previous paragraph, and found inhomogeneous distance distributions and partial activation. The coexistence of two structural states and an incomplete shift to one structural state upon Ca^2+^ activation have also been reported for tropomyosin positioned in cardiac muscle thin filaments visualized by cryoelectron microscopy [74].

Phosphorylation is important for the fine regulation of muscle tension and Ca^2+^ sensitivity. In contrast to skeletal TnI, cTnI possesses a unique *N*-extension (1–32 amino acids) that does not have a crystal structure [59]. This extension is functional for sufficient opening of the cTnC N-domain [75,76,77], as well as maintaining the normal Ca^2+^ sensitivity of cTnC [61,78,79,80]. Furthermore, two phosphorylation sites (amino acid residue numbers Ser23, Ser24 and Ser42, Ser44) for protein kinase A (PKA) and C (PKC), respectively, are present in or adjacent to this extension and modulate Ca^2+^ sensitivity after phosphorylation [81,82]. The structure and location of the N-terminal extension were studied by NMR, X-ray/neutron scattering, FRET, and PELDOR/DEER, but little is known because of its intrinsically disordered protein (IDP) nature [65,83,84,85,86]. Zhao et al. [87] determined the secondary structure of the N-extension of cTnI by measuring the distance distribution between spin labels attached to the *i* and *i +* 4 residues using CW-EPR (Figure 4). Two residual distance distributions exist, with the major distribution one spread over the range from 1 to 2.5 nm and the other minor peak at 0.9 nm. Only slight or non-obvious changes were observed when the extension was bound to cTnC in the cTnI–cTnC complex at physiological ionic strength (0.2 M KCl). However, at 0.1 M KCl, the distance between two residues, 43/47, located at the PKC phosphorylation sites (Ser42 and Ser44) on the boundary of the extension, exclusively exhibited the 0.9 nm peak, as expected from the α-helix in the crystal structure of the complex. Furthermore, the distance of Residues 23/27, located on the PKA phosphorylation sites Ser23 and Ser24, showed that the major distribution was markedly narrowed, centered at 1.4 and 0.5 nm wide, accompanying the spin-label immobilization of Residue 27. The results show that the extension exhibited a primarily partially folded or unfolded structure equilibrated with a transiently formed α-helical-like short structure over its length. It has been hypothesized that the specific secondary structure of the extension of cTnC becomes uncovered when the ionic strength descreases, demonstrating that only the phosphorylation regions of cTnI interact stereospecifically with cTnC. The 23–27 regions of cTnI bind to cTnC, which stabilizes the conformation with high Ca^2+^ affinity. Phosphorylation of Serines 23 and 24 abolishes this interaction, having an effect similar to truncation or removal of the N-extension of cTnI [78]. It is possible that upon phosphorylation, the introduction of negative charges and localized conformational changes within the N-terminal extension of cTnI reduces its affinity for cTnC, thus relieving the stabilization and enhancing Ca^2+^ release, as described below.

It is well known that the two adjacent serine residues at positions 23 and 24 of the cTnI extension are phosphorylated by PKA. This phosphorylation decreases the binding constant of cTnC for regulatory Ca^2+^ [78] to increase the dissociation rate of Ca^2+^ from cTnC and finally enhance the relaxation rate of the heart [78,88]. The interspin distances within the N-lobe of cTnC in the cTnC–cTnI complex showed small changes (<0.5 nm) observed by CW-EPR, demonstrating the full opening and exposure of hydrophobic patches upon Ca^2+^ addition [70]. Okawa et al. (unpublished [89]) and Somiya et al. (unpublished [90]) measured these inter-helix distances and the distance between the TnC N-lobe and the TnI switch region upon phosphorylation. The results suggested that the phosphorylation induced incomplete cTnC opening and cTnC–I tightness increased the dissociation rate of Ca^2+^ from cTnC and decreases the binding constant of cTnC for regulatory Ca^2+^, resulting in an enhanced relaxation rate of the heart.

It is also well known that the serine residue at position 206 of cTnT is phosphorylated by PKC. This phosphorylation decreases actomyosin ATPase activity and muscle tension [61,91] to finally reduce the ATP consumption of the hypertrophic heart for adaptation or rescue. The phosphorylation of cTnT is an important mechanism for the fine regulation of cardiac physiological output, which is thought to involve alterations in the Ca^2+^-induced structural transitions in the cTnT–tropomyosin ™ linkage on the thin filament [61]. The troponin complex in the thin filament becomes flexible by a disease-causing mutation of cTnT, as detected by X-ray/neutron scattering [66,67]. The N-terminal domain of skeletal TnT shows Ca-dependent movement, as detected by the FRET distance measurement [92]. Yamashita et al. (unpublished [93]) measured the distance between the N-terminal region of cTnT and the C-terminal region of Tm on a reconstituted thin filament. The distance was consistent with that in the crystal structure of the fragmented cTnT and Tm complex [94]. Sakai et al. (unpublished [95]) found that upon PKC phosphorylation, the distance became longer by ~1 nm, demonstrating that this region split from Tm. This suggests that the split N-terminal domain of cTnT interferes with myosin binding to actin.

As there is no high-resolution structure of the thin filament available, the molecular mechanism of this regulated process remains uncertain. A well-known steric blocking model in which the binding of Ca^2+^ to Tn triggers the movement of Tm to a different position on the actin filament [96,97,98] is based on X-ray diffraction and electron microscopy [74,99,100,101,102,103,104]. In the three-state model, the movement of Tm occurs upon transition from the blocked to the closed state and proceeds further upon binding of the myosin heads to actin, resulting in the open state [105,106]. To identify interaction sites, Ueda et al. [107] used the SDSL-EPR method and measured the rotational motion of a spin label covalently bound to the side chain of a cysteine genetically incorporated into rabbit skeletal muscle tropomyosin (Tm) at 10 positions along the entire length (Figure 5). Upon the addition of F-actin, the mobility of all the spin labels, especially at the position joint region of Tm, was significantly inhibited. Slow spin-label motion at the C-terminus was observed upon addition of troponin. It has also been suggested that the insensitivity of spin-label mobility resulted from no or small Ca^2+^-induced movements of Tm. This strongly supports previous studies that showed no Ca^2+^-induced changes in the side-chain mobility of the spin label at 36 and 190 [108], in the back-bone mobility of bifunctional spin label of Tm measured by saturation transfer EPR [14], or in the FRET between Tm and actin residues found by the Miki group [109,110,111].

To directly detect how Tm moves on actin in the thin filament, Ueda et al. (unpublished [112]) measured the distances between the a ^15^N-spin label on the residues of Tm and a ^14^N-spin label on the native cysteine residue 374. Tsujimoto et al. (unpublished [113]) also measured the distances between spin-labeled residues of Tm and the paramagnetic Mn^2+^–ADP of actin. Later, we succeeded in mapping the spin-labeled Tm residues on the actin filament using these two distances in the presence and absence of Ca^2+^. The results demonstrated a slight axial movement (<0.5 nm) of Tm toward the pointed end upon Ca^2+^ addition. Interestingly, the joint portion of the N-terminal and C-terminal regions of Tm loosened and flared slightly in the –Ca^2+^ states. This suggests that the average position of Tm moves slightly upon Ca addition. This may be consistent with the reports of multiple structural states and an incomplete shift to one structural state upon Ca^2+^ activation for the Tm position of cardiac muscle visualized by cryoelectron microscopy [74].

The binding of myosin-head S1 fragments without troponin immobilized Tm residues at positions along its length, suggesting that these residues are involved in a direct interaction between Tm and actin in its open state [107]. As immobilization occurred at substoichiometric amounts of S1 binding to actin (a 1:7 molar ratio), the structural changes induced by S1 binding to one actin subunit must have propagated and influenced interaction sites over seven actin subunits. To identify the interaction sites of Tm, Ueda et al. [114] measured the rotational motion of a spin-label covalently bound to the side chains of Tm throughout the entire length. Most of the Tm residues were immobilized on actin filaments with S1 bound. The residues in the midportion of Tm were mobilized when the troponin (Tn) complex was bound to the actin–Tm–S1 filaments. The addition of Ca^2+^ ions partially reversed the Tn-induced mobilization. In contrast, residues at the joint region of Tm were unchanged or oppositely changed. These results indicated that Tm was fixed on thin filaments with myosin bound, although a small change in the flexibility of the side chains of the Tm residues, presumably interfaced with Tn, actin, and myosin, was induced by the binding of Tn and Ca^2+^. These findings suggest that even in the myosin-bound (open) state, Ca^2+^ may regulate the contractile properties of actomyosin via Tm.

It has been suggested from the X-ray diffraction of skeletal muscle fibers that actin undergoes conformational changes during the contraction and regulation of Ca^2+^ [102]. Yamamoto et al. (unpublished [115]) measured the distance between a spin label of the native cysteine 374 and a paramagnetic Mn^2+^ ATP of actin by CW-EPR spectroscopy (Figure 6). After polymerization, the distance became shorter by 0.3 nm, and then actin transformed from the open to closed state. Myosin-S1 binding to filamentous actin returned the conformation back to the open state in a cooperative manner at substoichiometric amounts of S1 binding to actin (a 1:3 molar ratio), and back again toward the closed state at further increased amounts of S1. Interestingly, actin transformed to the open conformation in the reconstituted thin filament (actin–Tn–Tm complex) and went back to the closed state after S1 binding. However, Ca^2+^ had no effect on the actin conformation of the thin filament. It was therefore suggested that the actin monomer does not undergo large conformational changes, at least of nucleotide binding and cysteine 374 regions of actin molecule of the Ca^2+^-regulated thin filament, although it is well known that the Tn–Tm complex moves on the actin filament of the Ca^2+^-regulated thin filament. Ca^2+^-regulated movement of the Tn–Tm complex may occur on a surface of the actin monomer of the filament.

A more detailed 3D arrangement of actin and Tm in the thin filament will be explored using other sites, including the spin-labeled phalloidin binding site of actin (Nishi et al., unpublished [116]).

## 5. Kinesin ATPase

Kinesin ATPase is a microtubule (MT)-related motor protein [117,118]. ATP is hydrolyzed via the following reaction: MT.K + ATP → MT.K.ATP → MT.K.ADP.P → (MT)K.ADP + Pi → MT.K + ADP. The active movement of kinesins supports several cellular functions, including axonal transport in the nerve [119]. As shown in Figure 7A, kinesin has two heads (motor cores) that are smaller than myosin heads, with carboxy-terminal ends connected via two neck-linkers (NLs) that are both jointed to a coiled-coil stalk [120,121,122]. The binding sites of kinesin for MT include Loops 11 and 12 [123]. We used the SDSL-EPR technique to study the structural dynamics of kinesin. Upon MT binding and in the presence of nucleotides ADP and the ATP analog AMPPNP, the spin labels on L11 significantly decreased the fraction of the slow component. Moreover, dipolar CW-EPR detected a wide distribution in the distance range of 1–2 nm between the two spin labels attached to two amino acids separated by two–three residues. This distribution was slightly narrower in the presence of MTs than in their absence. The L11 residues underwent a conformational transition upon the binding of nucleotides and MT, while these residues continued to fluctuate over a nanometer range while kinesin bound to MT [124]. Using EPR and other methods, it has been established that NLs undergo a nucleotide-dependent undocked and docked equilibrium on the motor core [125,126]. Using monomeric kinesin without a coiled-coil stalk, Takai et al. (unpublished [127]) measured the distance between NL (spin label at Cys 336) and the motor core (spin label at Cys 223) (Figure 7A and Figure 8B), and showed a broad distance distribution (0.8–2 nm) in AMPPNP state but the ADP and no nucleotide states were beyond the sensitivity (>2.5 nm). Sugata et al. [128,129] spin-labeled each of the NLs at Cys 332 and measured their mobility and interspin distance on the kinesin–MT complex by CW-EPR and PELDOR/DEER spectroscopy (Figure 7B,C and Figure 8A). The results demonstrated that the NLs were docked (inter-NL length: 1.5–2 nm) and undocked (2–5 nm) on the motor core and always in equilibrium between two docked–undocked states at various nucleotide states. The inter-NL length of the docked state was longer with broader fluctuations in the no nucleotide state than in the ATP analog (ATPγS or AMPPNP) and ADP states. These results suggested that the NL is undocked almost totally in the no nucleotide state, whereas a docked–undocked equilibrium exists in the ATP analog and ADP states. It is possible that in the ADP state, NL docks to the core by interaction between the two motor cores, but not to monomeric kinesin. Takai et al. (unpublished [127]) demonstrated that the stalk and NL of kinesin work as a strain gauge. The stalk coiled-coil will melt under high stress. The distance was short only in the no nucleotide state, where there was no stress between the NLs. In the AMPPNP and ADP states, the distances were beyond the sensitivity of CW-EPR, suggesting that the strong stress will work for strong MT binding in the AMPPNP state and that thermal force will work in the ADP state by the dissociated motor domains. However, it was suggested that NLs are too flexible to generate force because an undocked state exists in all nucleotide states. A similar conclusion was also reached from the small free energy change associated with AMPPNP-induced docking of spin-labeled NL onto the catalytic core [125]. It has therefore been hypothesized that the positioning of the neck-linker determines the nucleotide state of the motor core, which ultimately determines its binding strength to MT. Kinesin motor core–ADP moves forward by thermal (Brownian) motion, and NL is then pulled backward (undocked) from the motor to release ADP, and the motor binds strongly to MT in the no-nucleotide state. The NL of the rear motor is pulled forward and docked to hydrolyze ATP to ADP and makes the motor core dissociate from MT.

How does the neck-linker positioning (docking) determine the nucleotide state? SDSL-EPR studies [130] suggest that two helices, Switch I and Switch II, undergo nucleotide-dependent conformational changes in G-proteins [131] that play an important role in communicating between the catalytic site and the NL docking domain on the MT-binding interface of kinesin. Recent cryoelectron microscopy (cryo-EM) studies have also revealed that these unique helices communicate between the catalytic site and the NL docking region [123,132,133]. In kinesin X-ray crystal structures, the N-terminal region of the α-1 helix is adjacent to the adenine ring of the bound adenine nucleotide, while the C-terminal region of the helix is near the NL [134]. Yasuda et al. [135] monitored the nucleotides using site-directed spin-labeling EPR (Figure 9). Kinesin was doubly spin-labeled at the α-1 and α-2 helices, and the resulting EPR spectrum showed dipolar broadening. The interhelical distance distribution showed that 20% of the spins had a peak characteristic of 1.4–1.7 nm separation, which was similar to what was predicted by the X-ray crystal structure, although 80% were beyond the sensitivity limit (>2.5 nm) of the method. Upon MT binding, the fractions of kinesin exhibiting an interhelical distance of 1.4–1.7 nm in the presence of AMPPNP and ADP were 20 and 25%, respectively. In the absence of nucleotides, this fraction increased to 40–50%. These nucleotide-induced changes in the fraction of kinesin undergoing displacement of the α-1 helix were found to be related to the fraction in which the NL was undocked from the motor core. It has been concluded that a shift in the equilibrium between two α-1 helix conformations occurs upon nucleotide binding and release, and that this shift controls NL docking to the motor core through the interaction of the C-terminal region of the α-1 helix with NL. We propose that this conformational transmission occurs in both directions, from the nucleotide site to NL and from NL to the nucleotide site (Figure 10). NL docking to the motor core then also controls the nucleotide state via a shift in the equilibrium between two α-1 helix conformations. For example, undocking of NL induces a α-1 helix shift towards an α-2 helix to accelerate ADP release at the catalytic ATPase site. Docking of NL induces α-1 helix replacement away from the α-2 helix to accelerate ATP hydrolysis and produce ADP at the ATPase site.

## 6. Clock ATPase

The clock or rhythm system is considered to be the energy transduction system that reduces entropy. The cyanobacterial clock proteins KaiA, KaiB, and KaiC interact with each other to generate circadian oscillations. KaiA enhances KaiC’s autophosphorylation [136,137,138,139] and ATPase activities [140,141], whereas KaiB reduces the phosphorylation level of KaiC enhanced by KaiA [142] and suppresses KaiC ATPase activity [140]. The circadian clock machinery, which consists of these three clock proteins (KaiA, KaiB, and KaiC) can be reconstituted in vitro by mixing the three Kai proteins in the presence of ATP; the phosphorylation level [143] and ATPase activity [140] of KaiC and the formation of the complexes between KaiA, KaiB, and KaiC [144,145] oscillate with a period of approximately 24 h in the in vitro reconstituted clock system.

In collaboration with the Ishiura group, Ishii et al. [146] identified the residues of the KaiA homodimer affected by the association with hexameric KaiC (KaiC6mer) using a spin-labeled KaiA C-terminal domain protein (KaiAc) and performing SDSL CW-EPR analysis. The distance between the two residues, located on the mobile lobe of the KaiA dimer, decreased after KaiC association (Figure 11B,C), suggesting that the two mobile lobes approach each other and clamp onto KaiC during the interaction (Figure 11A). Ishii et al. (unpublished [147]) also identified the residues of the KaiC6mer affected by association with the KaiA dimer using a spin-labeled KaiC C-terminal tail domain. The average distance between the six residues located on the mobile tails of the KaiC6mer increased with a broader distance distribution after KaiA association (Figure 11D–G), suggesting that the six tails of KaiC become pulled out, extended, separated from each other, and disordered during the interaction of KaiC with KaiA (Figure 11A). Only a small fraction of the tails exposed or leaped out thermally in the absence of KaiA, because the inter-tail distance distributions were narrower in the absence of KaiA than in its presence (Figure 11F,G). The fraction of extended tails increased when the exposed tails captured KaiA and did not retract back into the KaiC cavity. (Alternatively, KaiA may pull out the tail directly.) This view supports an attractive possibility that the extension or leaping-out of the tail transmits into the inside cavity of the KaiC6mer through loop and reaches autophosphorylation and ATPase sites to promote autophosphorylation and ADP release [148,149] (see Figure 11A).

## 7. Ion Motive ATPase

The P-type ATPases, also known as the E1–E2 ATPases, are a large group of ion pumps that are found in bacteria, archaea, and eukaryotes. Examples of P-type ATPases are the sodium–potassium pumps (Na^+^/K^+^–ATPase), the proton–potassium pumps (H^+^/K^+^–ATPase), the calcium pumps (Ca^2+^–ATPase), and the plasma membrane proton pumps (H^+^–ATPase) of plants and fungi [150,151]. The sarcoplasmic calcium pump (Ca^2+^–ATPase) was first crystallographically studied using ATP and phosphate analogs. The conformation of Ca^2+^ binding sites in the transmembrane domain is coupled with the movements of the three cytoplasmic domains driven by ATP hydrolysis reaction (see Figure 12 by replacing Cu for Ca). The transition of intermediates from E1-P.Ca_2_ to E2-P + Ca^2+^ is generally accepted as a key reaction. To examine whether this ATPase has a similar structure in the crystal and membrane structures, Narumi et al. [152] identified 15 residues from the surface of the sarcoplasmic reticulum Ca^2+^-pump ATPase by mass spectrometry using diethylpyrocarbonate sites, and found that it is composed of 10 membrane-spanning segments (M1–10) [150]. The N- and P-domains possess the nucleotide-binding site and the phosphorylation site, respectively. The A- or actuator domain plays a significant role in converting the large-scale conformational changes of the cytoplasmic domains and those of the membrane domain through its rotation during the reaction cycle (see Figure 12, replacing Ca for Cu). The coupling of the organization and large-scale movements of the three cytoplasmic domains with the conformation of this membrane domain induces the key events in this reaction cycle. The transition of intermediates from E1PCa^2+^ to E2P + Ca^2+^ is accepted as a key reaction. In order to examine whether the ATPase has similar structures in crystalline and membrane forms, we identified 15 residues from the surface of sarcoplasmic reticulum Ca^2+^-pump ATPase by mass spectrometry using diethylpyrocarbonate modification. Most residues are well explained by solvent accessibility, pKa, and nearby hydrophobicity of the reactive atom on the basis of the atomic structure, strongly suggesting that this ATPase undergoes similar structural changes upon substrate binding in the crystal and in the membrane. However, the reactivity of four residues near the interface among the A-, N-, and P-domains suggests larger conformational changes of these domains in solution than in the crystal upon binding of Ca^2+^, ATP, and MgF_4_, suggesting that these domains move dynamically in solution. It is considered that ATPases work as monomers or dimers in the membrane [153,154]. The spatial arrangement of rhodopsin could be examined in the native membrane with and without light irradiation by using spin-labeling and DEER/PELDOR [155]. This may be applicable to P-type ATPases in the membrane.

The coordination bond structure of transport ions can be studied by CW-EPR using paramagnetic ions and heavy metal transporters. Kuwabara et al. [157] isolated bacterial copper transporter CopB from *Thermus thermophilus,* and Daimon et al. (unpublished [158]) found that the EPR spectrum of Cu(II) bound to copper transporter CopB showed the typical characteristics of type II copper [159,160]. Interestingly, Yasuda et al. [156] demonstrated that the spectrum is broader in the absence of nucleotide than in the presence of a nonhydrolyzable ATP analog (AMP-PCP), suggesting that the coordination structure of E1Cu_2_ has inhomogeneous or multiple conformations compared to that of E1.Cu_2_.ATP (see the spectra in Figure 12, inset). In other words, E1.Cu_2_.ATP has an occluded conformation, while E1.Cu_2_ has a loose conformation.

How is the ion transport coupled to the ATPase reaction and structural change from E1 to E2? Recent studies have reached the conclusion that the structural transition from E1-P.Ca^2+^ to E2-P.Ca^2+^ precedes Ca^2+^ release [161,162]. The FRET from the fluorescently labeled cytoplasmic domains A and P of the *Listeria* Ca pump directly shows that the structural transition (thermal large-scale domain movement) from E1 to E2 is faster than Ca^2+^ release [162].

## 8. Principle for Energy-Transducing Mechanisms

Some considerations or theories for biological chemomechanical conversion have been published [163,164,165,166,167,168,169,170,171]. For most of the theories [163,164,165,166,167,168,169,170], the chemical reaction or ligand binding governs and precedes or simultaneously occurs with the output structural change (Figure 13a,b). This is now known as induced-fit mechanism for enzymes or the power-stroke mechanism for motor proteins. On the other hand, it is now established that the chemical state of an enzyme only modulates the equilibrium between multiple structural states, and that the product (e.g., ADP or Pi) released from an enzyme confers irreversibility to the cycle. The second one is called the conformational selection mechanism for enzymes [172] or the Brownian ratchet mechanism for motor proteins. In order to answer which type of transducing mechanism is correct, we considered the relationships or common principles among the molecular mechanisms of various energy-transducing systems. First, we summarized all of our results and the evidence for the various systems.

(1) Myosin ATPase. The light-chain-binding domain (lever arm) is a mobile or flexible element. The thermal angular fluctuation of the lever arm drives the structural change (isomerization) of AM.ADP.P (non-force) to AM.ADP.P (force) to generate force, and the latter promotes subsequent phosphate release, which prevents the backward structural change.

(2) Actin–Tn–Tm filament. The TnI C-terminal domain (regulatory and switch domain) is a flexible element. Thermal fluctuation of the TnI switch domain occurs between the inactivated and activated state, even in the −Ca^2+^ state, and the latter state is stabilized by the binding of Ca^2+^ to cTnC. Tm laterally or radially fluctuates on actin filaments. The average position of Tm on actin filaments shifts slightly upon Ca^2+^ activation, suggesting multiple structural states and incomplete shifts to one structural state.

(3) Kinesin ATPase. Kinesin has two NLs that are flexible and mobile elements. Kinesin motor core–ADP moves forward by thermal motion, and then NL is pulled backward (undocked) from the motor to release ADP, followed by strong motor binding to MT in the no nucleotide state. The NL of the rear motor is pulled forward and docked on to hydrolyze ATP to ADP and makes the motor core dissociate from MT.

(4) Clock ATPase. KaiC6mer has six carboxy-terminal tails that are flexible and mobile elements. Only a small fraction of the tails may expose or leap out thermally in the absence of KaiA. The fraction of extended tails increases when the exposed tails capture KaiA, extend, and do not retract back into the KaiC cavity. The extension or leaping-out of the tail transmits into the inside cavity of the KaiC6mer through the loop and reaches autophosphorylation and ATPase sites to promote autophosphorylation and ADP release.

(5) Ion motive pump ATPase. The P-type ATPase has cytoplasmic domains that fluctuate largely with each other, pulling a transmembrane helix that gates the transport ion(s) to release outside. The structural thermal transition of the cytoplasmic domains and the transmembrane helix from E1-P.Ca_2_ to E2-P.Ca_2_ precedes Ca^2+^ release.

Altogether, all these systems have a flexible or mobile element with at least two reversible states in thermal equilibrium. Large fluctuations are attained by these elements. This is good evidence for the conformational selection or Brownian ratchet mechanism, because it is suggested that if the motor undergoes small fluctuations, the ratchet mechanism will not be indistinguishable from the power stroke mechanism [173]. For the switching mechanism, the second state is stabilized by the binding of the ligand (Figure 13c). For energy transduction, the thermal large-scale structural transition of this element from one state to the other precedes the subsequent irreversible chemical reaction, which stabilizes the latter state and prevents the reversal of the structural change (Figure 13d).

## Figures and Tables

**Figure 1 ijms-21-00672-f001:**
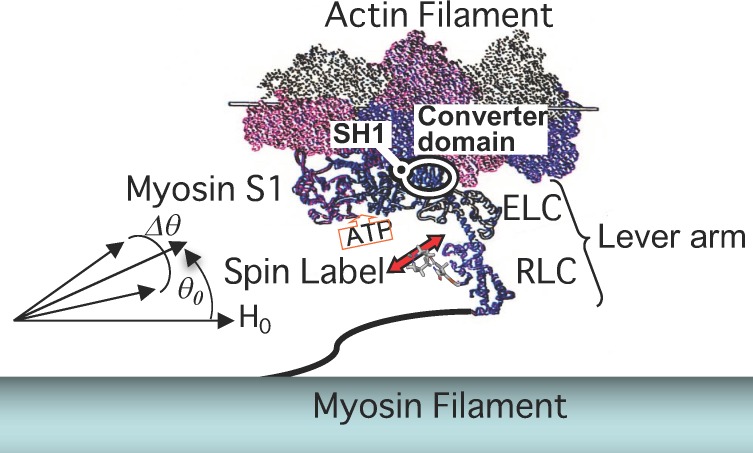
Structural model of a myosin head bound to an actin filament (actin monomers shown in different colors) in a muscle fiber. The spin label is attached to SH1 (Cys707) of the myosin converter domain and Cys154 of myosin RLC. The mobility of the SH1 spin label was monitored with the muscle fibers perpendicular to the applied magnetic field. The orientation of the RLC was determined as an angle (center θ_0_ and full width ∆θ of Gaussian distribution) of the principal axis (red bidirectional arrow) in the respect to the long axis of the muscle fiber parallel to the H_0_ field.

**Figure 2 ijms-21-00672-f002:**
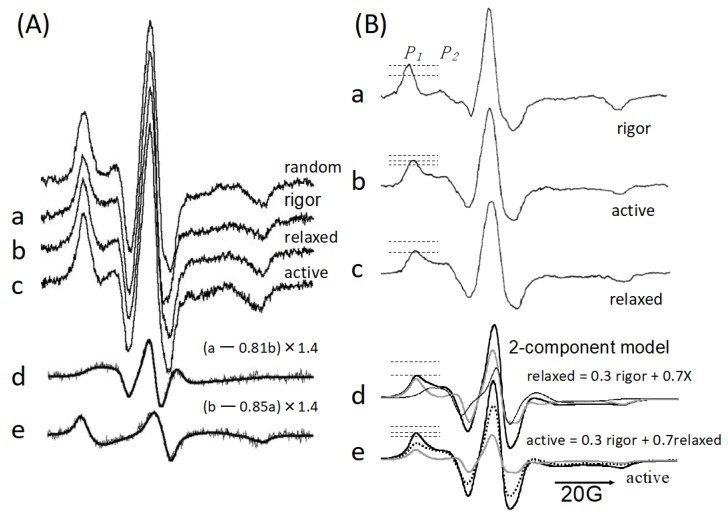
Electron paramagnetic resonance (EPR) spectra from the spin-labeled myosin head in muscle fibers in the rigor (a), relaxed (b), and isometrically active (c) states. The spin label was attached to Cys154 of the myosin RLC (**A**) and SH1 (Cys707) of the myosin converter domain (**B**). The fibers were inserted into a glass capillary that was perfused with a solution containing ATP [4]. (**A**) Orientation analyses were performed for the fibers parallel to the applied magnetic field (Arata, unpublished [37,38,39]). The difference between the rigor and relaxed spectra demonstrated two kinds of well-ordered spectra with narrow linewidth (d,e). From the decomposition analysis, the spectra a–c were composed of these two distinct angular populations: a = 0.60d + 0.40e, b = 0.50d + 0.50e, c = 0.55d + 0.45e. The fits were improved by incorporating random spectra: a = 0.40d’ + 0.20e’ + 0.40random, b = 0.30d’ + 0.30e’ + 0.40random, c = 0.32d’ + 0.28e’ + 0.4random. d’ and e’ had slightly narrower linewidths than those of d and e, respectively. (**B**) The mobility of the SH1 spin label was monitored with the muscle fibers perpendicular to the magnetic field [47]. Vertical dotted lines upper, middle, and lower at spectra a–e show the peak heights of lower magnetic field in rigor, active and relaxed state, respectively. The relaxed spectrum (thick black curve) was represented by two components, rigor (gray curve: 30%) and X spectrum (thin black curve: 70%). The active spectrum (thick black curve) was further composed of rigor (gray curve: 30%) and relaxed (dotted curve: 70%) populations.

**Figure 3 ijms-21-00672-f003:**
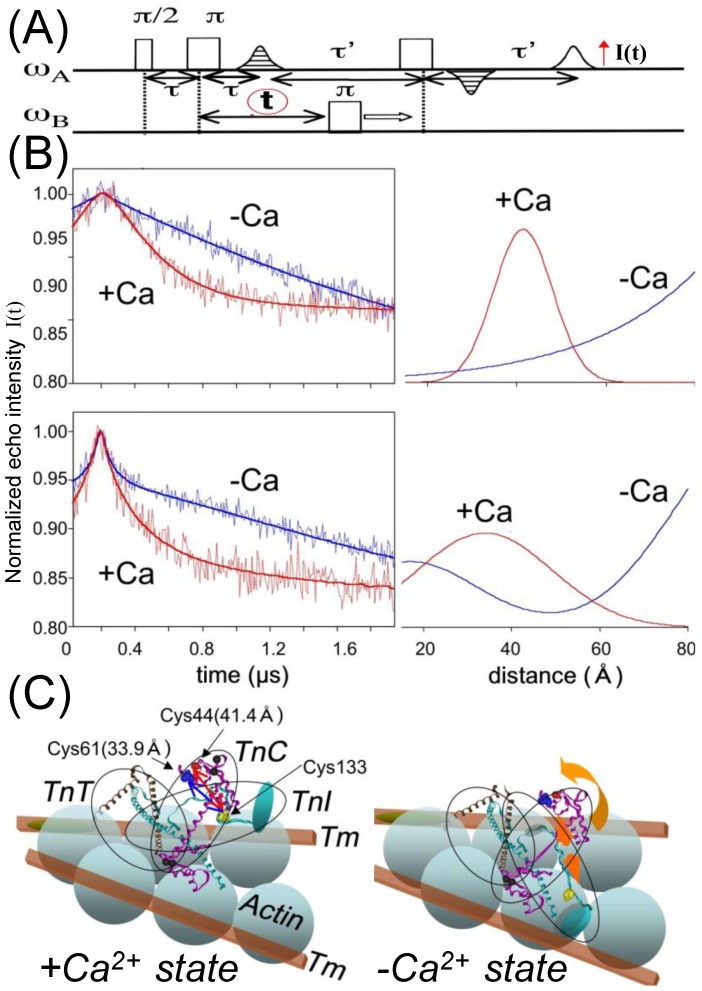
Analysis of the PELDOR/DEER spectra of interacting spins between troponin C and the troponin I switch region in the reconstituted thin filament. (**A**) Four-pulse sequence. The 90° pulse, the 180° pulse and ELDOR 180° pulse were applied. The ω_A_ and ω_B_ describe the resonance frequencies of the two spin species. The pulse separation τ and τ_0_ were set constant, and t was changed as shown by arrow [65,68]. Two peaks show the echoes appeared by this pulse sequence. The second echo intensity I(t) oscillates by changing t. (**B**) The spectra I(t) vs. t (**left**) and distance distributions (**right**) in the presence (red line) and absence (blue line) of Ca^2+^ ions. Spin pairs were between Cys133 in TnI and two different residues in TnC: Cys44 (**B, upper**) and Cys61 (**B, lower**). (**C**) The conformations of TnC and TnI in the heterotrimer of the thin filament. The structures of the ternary TnC–TnI–TnT2 complexes in the presence and absence of Ca^2+^ ions have been modified from PDB codes 1YTZ and 1YVO, respectively. Adapted from Aihara et al. 2010 [68] with permission from American Society for Biochemistry & Molecular Biology.

**Figure 4 ijms-21-00672-f004:**
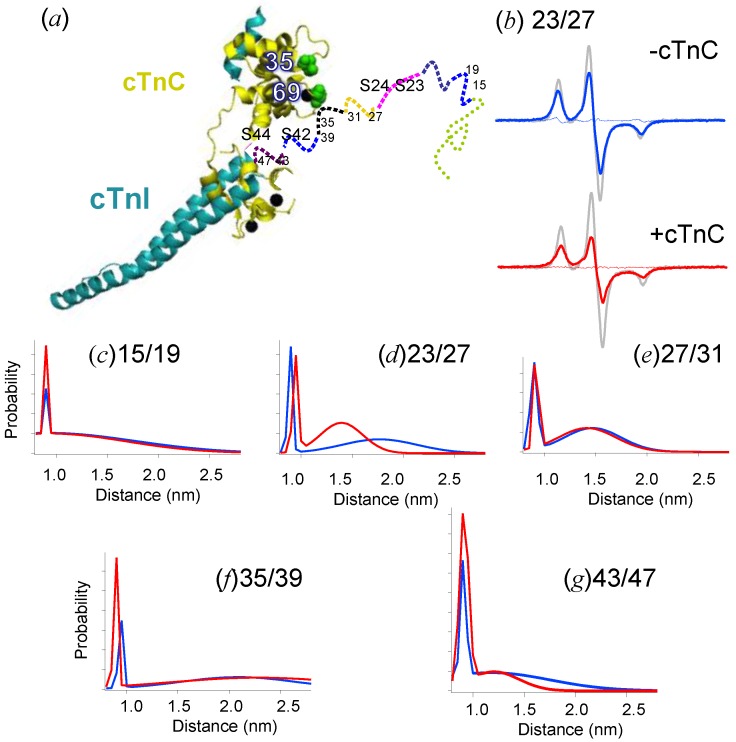
Distance analysis of the EPR spectra from the double MTSL-labeled cTnI mutant with and without cTnC at low ionic strength (0.1 M KCl) [87]. (**a**) Structure of the N-extension of cTnI in the dimeric complex with cTnC. (**b**) The continuous wave EPR spectra from the cTnI mutant labeled at Residues 23/27 in the absence (blue line) and presence (red line) of cTnC. The spectra were obtained at 170 K. The scan width was 200 G. The experimental spectrum (colored line) was compared with the spectrum of the single-labeled mutant (gray line). The experimental spectra with or without cTnC were fitted by two Gaussian distance distributions shown by the red or blue line in (**d**), respectively. Residuals from the best-fit spectrum are shown as the thin line. (**c**–**g**) Two Gaussian distance distributions shown by the red and blue lines were obtained from cTnI mutants labeled at Residues 15/19 (**c**), 23/27 (**d**), 27/31 (**e**), 35/39 (**f**), and 43/47 (**g**) with or without cTnC, respectively. Adapted from Zhao et al. 2019 [87].

**Figure 5 ijms-21-00672-f005:**
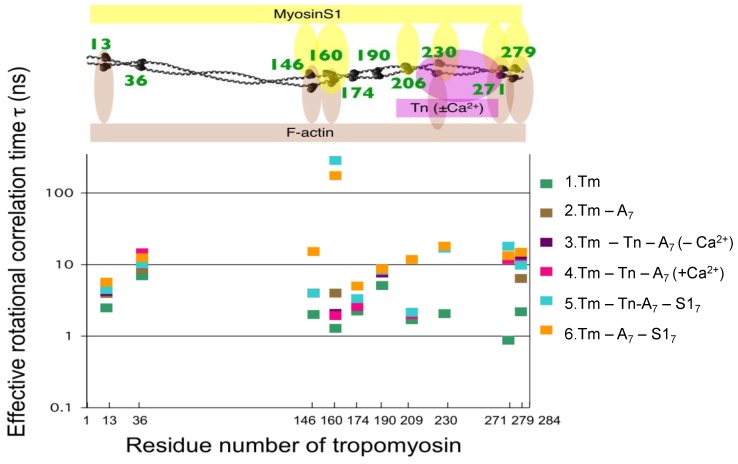
Profile of effective rotational correlation time as a function of the number of spin-labeled tropomyosin residues [107]. Each marker color indicates the sample preparation shown on the right side of the figure. When the spectrum consisted of fast and slow components, the major peak was used to estimate the effective rotational correlation time. The upper panel shows a model for the interaction of tropomyosin residues with actin and myosin S1. Note that Ca^2+^ had no effect on the tropomyosin site in the thin filament (Tm–Tn–A_7_).

**Figure 6 ijms-21-00672-f006:**
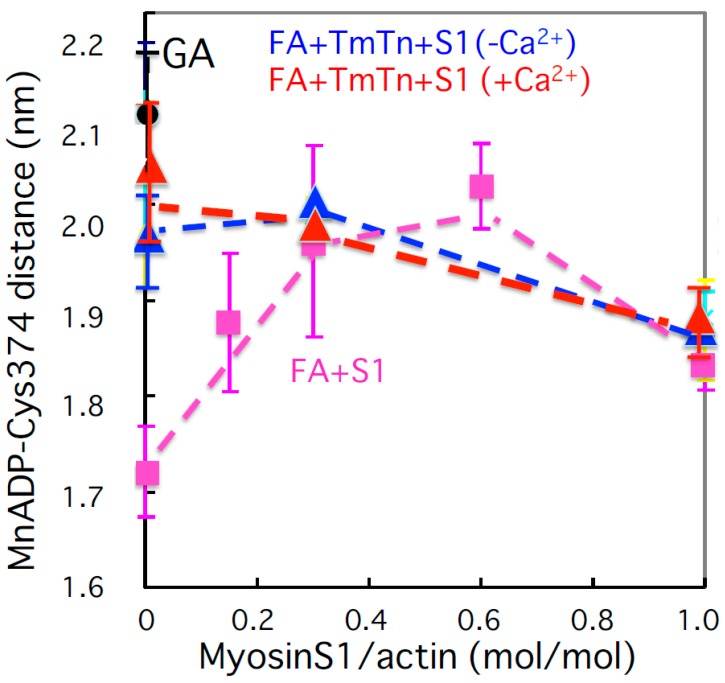
Distance between the spin label of the native cysteine 374 and paramagnetic Mn^2+^ATP of the actin filament as a function of molar ratio of bound myosin-S1 to actin (Yamamoto et al., unpublished [115]). The black circle indicates the distance of G-actin (GA) and the pink square shows the distance of actin filament alone (FA). Red and blue triangles show the distance of the actin in the thin filament (FA–Tm–Tn) with and without Ca^2+^, respectively.

**Figure 7 ijms-21-00672-f007:**
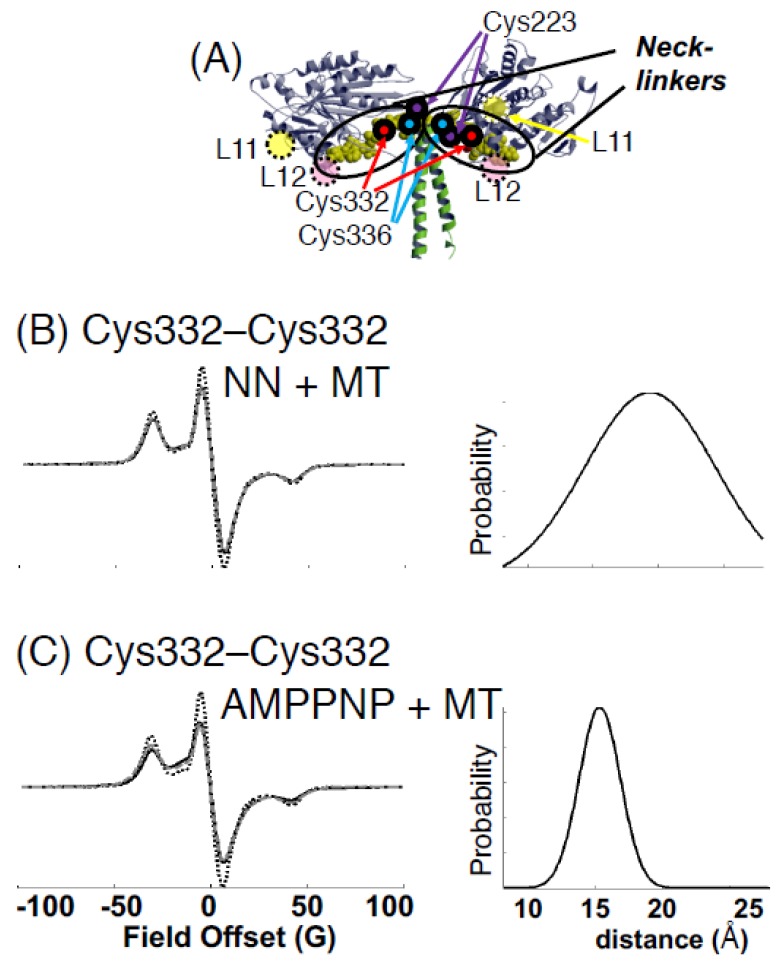
Analysis of the dipolar CW-EPR spectra of interacting spin pairs in kinesin frozen in solution. (**A**) Positions of the spin labels in the kinesin dimer. (**B**,**C**): EPR spectra for the kinesin dimer Cys332–MSL were taken at 150 K in the absence (**B**, NN + MT) and the presence (**C**, AMPPNP + MT) of AMPPNP with microtubules (MTs). For each state, the left panel compares the experimental spectrum of the doubly labeled kinesin dimer (continuous black line) with the spectrum of the corresponding monomer (dashed line), normalized to the same number of spins. Only a single Gaussian (**right**) satisfactorily fitted the experimental spectra. The best-fit spectra (continuous gray line) are shown on the left. Adapted from Sugata et al. 2009 [129] with permission from Elsevier.

**Figure 8 ijms-21-00672-f008:**
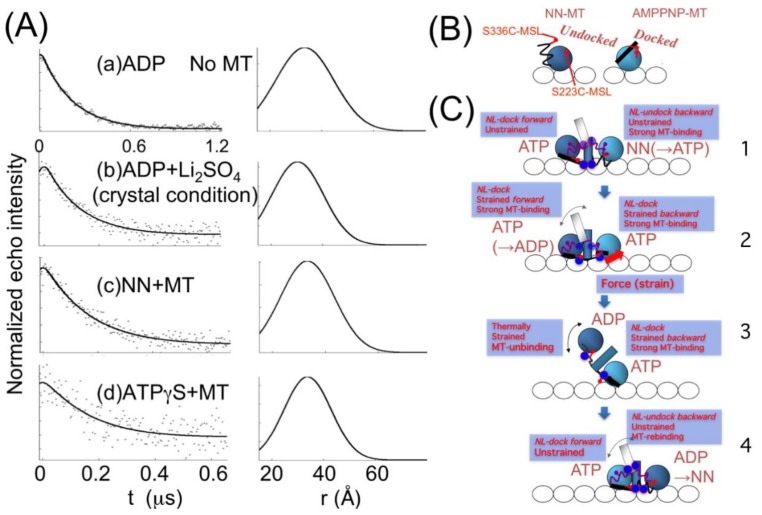
Analysis of the PELDOR/DEER spectra of interacting spin pairs of kinesin frozen in solution. (**A**) The spectra for the kinesin dimer Cys332–MSL (see Figure 7A) were taken with a four-pulse sequence in several nucleotide conditions, both in the absence and in the presence of microtubules (MTs) at 65 K. (**a**) ADP; (**b**) ADP + Li_2_SO_4_; (**c**) NN + MT; (**d**) ATPγS + MT. For each state, the left panel compares the experimental spectrum modulation of the doubly labeled kinesin dimer (points) with the fitted spectrum modulation (continuous line). Only the single Gaussian distribution shown in the right satisfactorily fit the experimental spectra. (**B**) Models for neck-linker (NL) docking of monomeric kinesin. (**C**) Models for the NL in dimeric kinesin during ATP hydrolysis. The flexible NLs work as a strain sensor rather than a force generator. Adapted from Sugata et al. 2009 [129] with permission from Elsevier.

**Figure 9 ijms-21-00672-f009:**
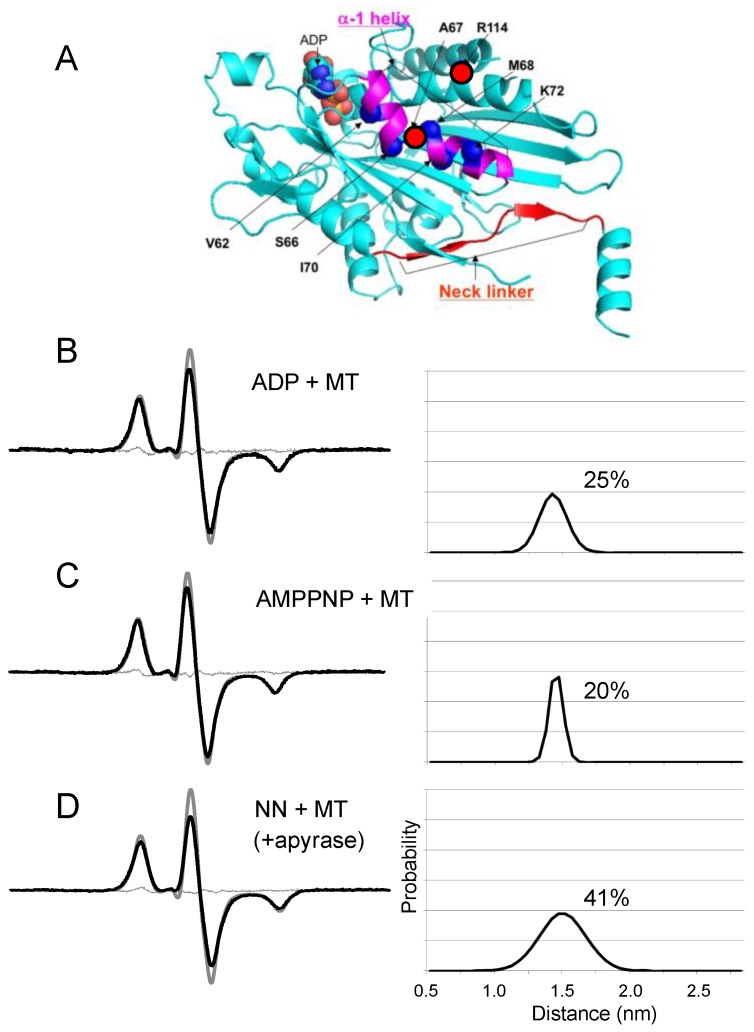
Distance analysis of the EPR spectra from the doubly labeled K68C/R114C kinesin mutant. The EPR spectra were obtained at 173 K under several nucleotide conditions, with or without MTs. (**A**) Positions of the spin labels on monomeric knesin (PDB code: 2KIN). The α-1 helix and NL are shown by purple spiral and red line and arrows (β-sheets), respectively. (**B**) The ADP-bound state in the presence of MTs. (**C**) The AMPPNP-bound state in the presence of MTs. (**D**) The apo or no nucleotide (NN) state in the presence of MTs. The scan width was 200 G. For each state, the left panel compares the experimental spectrum of the doubly labeled kinesin mutant (black line) with the spectrum of the single-labeled mutant (thick gray line) normalized to the same number of spins. The experimental spectra were fitted satisfactorily by a single Gaussian distance distribution (right panel) with some fraction of spins occurring beyond the sensitivity limit (>2.5 nm). The percentages shown are the fraction of the total spins with an interhelical distance characterized by the single Gaussian distribution: 100 × (1 − fraction of spins > 2.5 nm apart). Residuals from the best-fit spectra (thin gray lines) are shown in the left panel. Adapted from Yasuda et al. 2014 [135] with permission from Elsevier.

**Figure 10 ijms-21-00672-f010:**
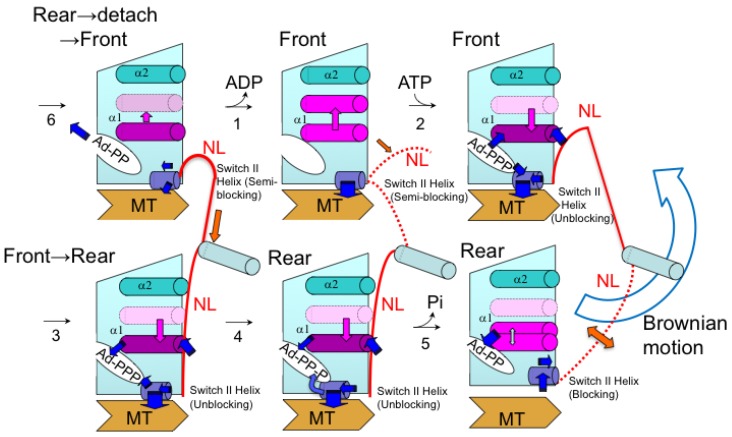
Proposed conformations of the α-1 helix in the ATPase intermediate states of dimeric kinesin on successive two tubulins. Protein–protein and protein–nucleotide interactions are shown as blue arrows. ATP (Ad-PPP) is hydrolyzed differently by the front and rear motor domains. Two successive tubulins are designated by two MTs aligned upright. The equilibrium shift of the neck-linker (NL) binding and α-1 helix position are represented by the orange and purple/violet arrows, respectively. Step 1: The NL of the front motor dissociates (as indicated by the orange arrow and the dotted NL) and is pulled backward and induces a shift in the positional equilibrium of the α-1 helix, bringing it closer to the α-2 helix by dissociation of the C-terminal region of the α-1 helix from NL. This results in the dissociation of the adenine ring of ADP to the N-terminal end of the α-1 helix. Step 2: A new ATP then binds to the front motor domain. The adenine ring binds to the N-terminal end of the α-1 helix. Switch II binds to the phosphate region and unblocks NL binding. Step 3: The C-terminal region of the α-1 helix binds to NL after NL is pulled forward from the rear motor domain when the other motor domain moves to the next site of MT and becomes the front motor domain. Step 4: Switch II of the rear motor domain accelerates ATP hydrolysis at the catalytic site. Step 5: Phosphate release from the rear motor domain. Step 6: The NL becomes bound weakly (indicated by bidirectional orange arrow). The rear ADP-bound motor domain dissociates from MT and searches for the forward binding site by Brownian motion. Adapted from Yasuda et al. 2014 [135] with permission from Elsevier.

**Figure 11 ijms-21-00672-f011:**
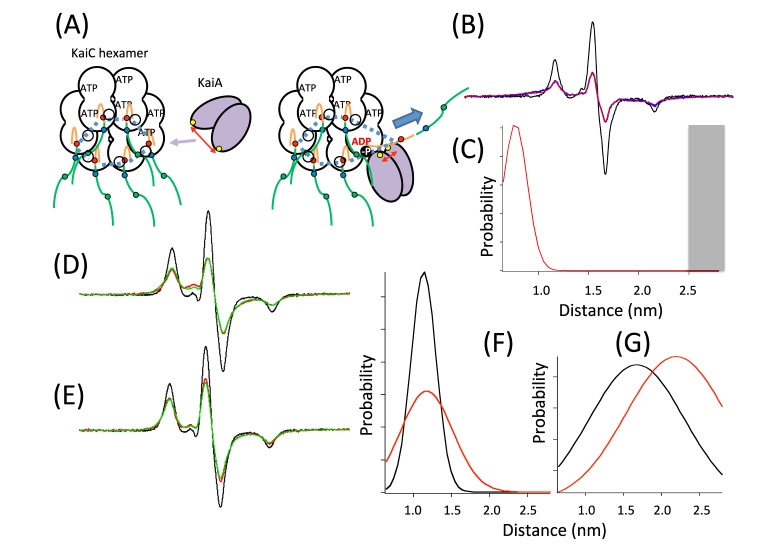
Analysis of the dipolar CW-EPR spectra of interacting spins in the KaiA dimer and KaiA–KaiC hexamer complex in frozen solution. (**A**) Positions of the spin labels in KaiA (yellow ball) and KaiC hexamer tails (red, blue, and green balls) and a proposed model for the KaiA–KaiC hexamer complex where the KaiC tail is exposed by its binding with KaiA and autophosphorylation sites (unphosphorlated or phosphorylated indicated by open or dark circle, respectively), and ATPase sites (indicated as ATP/ADP) are activated inside the KaiC cavity. Red arrows shows the distance between KaiA monomers with and without KaiC tail. Light purple arrow shows the binding of KaiA to KaiC tail. The thick dotted lines indicate distances between spin labels (red balls). The thick blue arrow indicates the leap-out or extension of KaiC tail. (**B**) EPR spectra for the KaiA dimer were taken in the absence (gray line) and presence (blue line) of the KaiC hexamer (Ishii et al., unpublished [147]). In panel (**C**) [146], a single Gaussian distance distribution satisfactorily fits the experimental spectrum (red line in (**B**)), while the distance in the absence of KaiC (gray line in (**B**)) was beyond the sensitivity (>2.5 nm) (gray column in panel (**C**). Panels (**D**) and (**E**) show the spectra for the proximal tail region (red ball in (**A**)) of the KaiC hexamer in the absence and presence of the KaiA dimer, respectively. The spectra were taken for the KaiC monomer (gray line) and hexamer (red line). Panel (**F**) shows single inter-tail Gaussian distance distributions (gray and red lines) in the absence and presence of the KaiA dimer, respectively. These distributions satisfactorily fit the experimental spectra of the proximal tail region (green lines in (**D**) and (**E**)). Panel (**G**) shows single inter-tail Gaussian distance distributions for the distal tail region of the KaiC hexamer (green ball in (**A**)) in the absence (gray line) and presence (red line) of the KaiA dimer.

**Figure 12 ijms-21-00672-f012:**
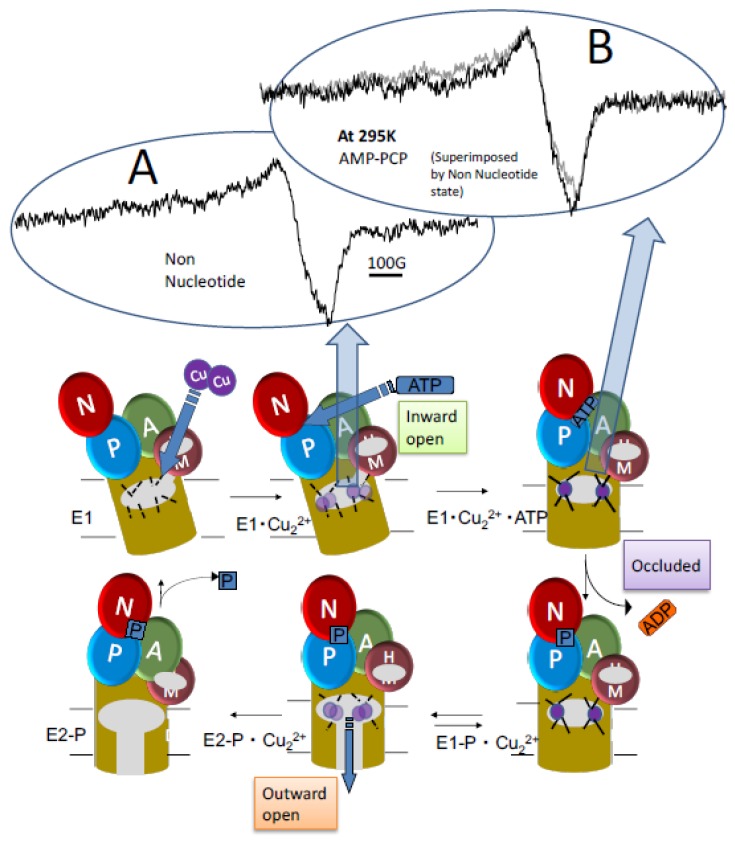
Mechanism for copper-transporting P-type ATPase. (**Upper panel** (inset A and B)): EPR spectra from Cu(II)^2+^ bound to transporter protein CopB at room temperature (Yasuda et al., unpublished [156]). The scan width was 1000 G. Note that the spectrum in the no nucleotide state (**left**) is broader than that in the ATP state (**right**) when the two spectra are superimposed on the right (gray line: nonucleotide state). It is proposed that more variable coordination structures are involved in copper binding in the E1Cu^2+^ state (loose) than in E1.Cu^2+^.ATP state (occluded).

**Figure 13 ijms-21-00672-f013:**
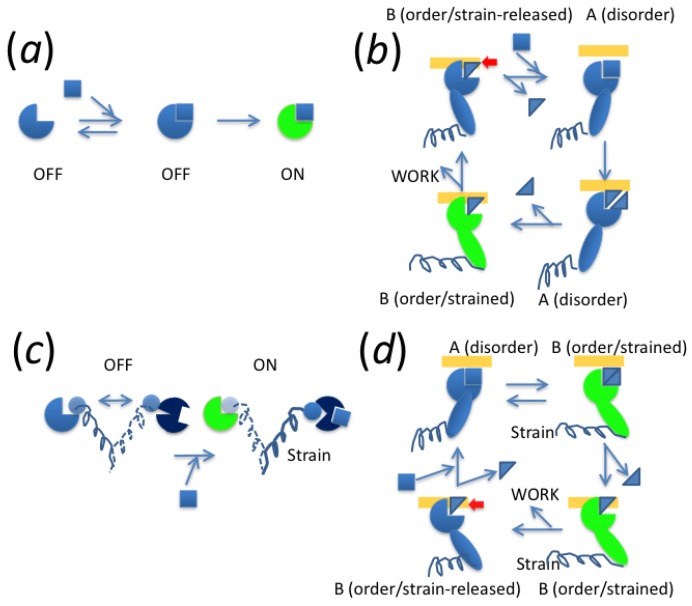
Principles for the energy-transducing and switching mechanisms. (**a**) Simple induced-fit switch. Ligand binding precedes or occurs simultaneously with a structural change. (**b**) Power-stroke mechanism. Product release precedes force generation. (**c**) Conformational selection switch. Structural equilibrium shifts to one side by ligand binding. (**d**) Brownian ratchet mechanism. Force generation precedes product release. Work is liberated by movement (red arrow). Wave lines indicate flexible element in c or spring-like structures in b and d.

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
