# Peer review of "Myosin and Other Energy-Transducing ATPases: Structural Dynamics Studied by Electron Paramagnetic Resonance"

_ijms, 2020, doi:10.3390/ijms21020672_

Round 1

Reviewer 1 Report

The author made extensive revision on the review on EPR studies on several energy-transducing protein machineries. I would say the revised version is much better than the original, but I believe that there are a number of relatively minor problems that need to be addressed before publication.

In addition, although the author corrected grammatical errors and typos that I pointed out, there still remain a few. Moreover, some expressions are awkward, and I encourage the author to have a native speaker go over the entire manuscript.

My original comment: (i) “It is an attractive possibility that efficient thermal motion will realize a long step beyond several actin monomers along actin filament in a fine organized sarcomere of myofibrils” (line 116 to 119).

Author response: I described more details on lines 222-228.
Under no load, thermal fluctuation of a weakly actin-bound (A)M.ADP.P (non-force) may be uncoupled with ATP hydrolysis because the chemical reaction (ADP and Pi release) is slow as compared with shortening rate of myofibrils. The large fluctuation and actin-myosin dissociation-reassociation cycles of (A)M.ADP.P (non-force) state may occur over several actin monomers. Only when forward stereospecific movement occurs, ADP and Pi releases occur from (A)M.ADP.P(force) to prevent backward movement.

My new comment: Please define “force” and “non-force” in parenthesis. 

Also, please explain how the speed of chemical reaction can be compared with the shortening rate of myofibrils. The unit of the former is /s, whereas the latter is µm/s, and therefore those two numbers cannot be directly compared.

Please also define “stereospecific movement” and explain why stereospecific movement cannot occur in the backward direction in this model.

My original comment: (ii) “the actin may undergo conformational changes for contraction but not for regulation” (line 344 to 345).

Author response: I described details but weakly on lines 447-452. Our suggestion is that actin monomer does not undergo large conformational changes of the Ca2+-regulated thin filament. Ca2+-regulated movement of Tn-Tm complex may occur on a surface of actin monomer of the filament.

My new comment: I was unable to find the relevant information in lines 447-452, but am I right to assume that the author retracted the original provocative statement?

My original comment:  (v) Expressions such as (23/24 and 42/44) (line 199) needs explanations.

Author response: I put explanations for 23/24 and residues 23/27. They indicate the phosphorylation site and the distance between residues 23 and 27, respectively, on line 300.

My new comment: Something is wrong with the line numbers, and I cannot find the relevant explanations in line 300. Nonetheless, if 23/24 could mean either two residues or the distance between the two residues, that is rather confusing. In fact, “residues 23/27, located on the PKA phosphorylation sites Ser23/24, showed that the major distance distribution was markedly narrowed” (line 267) does not make sense to me. 

In addition, I found several minor problems in the newly added sections.

(i) Please define the abbreviations in the first appearance.  “CW” in line 40 is not defined. In contrary, some are doubly defined. Examples are FRET, SDSL and DQC.

(ii) Please explain “EPR suppress(es) the relative orientation selection for nitroxide probes” (line 68).

(iii) “This method is very useful for the arrangement of a-helices within protein domain” (line 72). “useful in elucidating the arrangement of…”?

(iv) Please explain “We believe that the observed changes both in the distance within 1.5 nm … arise from the backbone movement or rotation, and/or…” (line 92).

(v) “More recently, Arata et al. (unpublished [20]), using X-ray scattering in collaboration with the K. Wakabayashi group, proposed oppositely bent structures as new intermediate states of ATP hydrolysis, similar to scallop ADP state [19]” (line 104-106). If the structure that Arata et al. proposed, based on a low-resolution method, is similar to a previously resolved atomic structure of S1, how can they claim that it is a new intermediate? Speaking of an oppositely bent structure, let me point out that Kimori et al. (2013) also reported such a structure (Biochem J, 450:23-35).

(vi) “The central goal in muscle energy transduction has been to detect the force-generating structural changes in…” (line 136).  “goal in muscle energy transduction research has been…”?

(v) “Abe et al. [69] also determined the large-scale distance between the cTnC N-lobe and the C-lobe by DQC in the reconstituted cTnC-cTnI complex” (line 226-228). What is “large-scale distance”?

Author Response

Thanks for the helpful comments for my manuscript. Following the reviewer's comments, I have addressed all the reviewer's comments in order and, for each answer, I detail our changes made to the manuscript according to the reviewers’ comments.  I also highlighted them in the text of revised version. I hope that this will be acceptable for publication in the International Journal of Molecular Sciences.

Response to Reviewer 1 Comments

Point 1: The author made extensive revision on the review on EPR studies on several energy-transducing protein machineries. I would say the revised version is much better than the original, but I believe that there are a number of relatively minor problems that need to be addressed before publication. In addition, although the author corrected grammatical errors and typos that I pointed out, there still remain a few. Moreover, some expressions are awkward, and I encourage the author to have a native speaker go over the entire manuscript.

Response 1: I will ask the editorial office to send the manuscript to MDPI English Editing Service after submission of revised version, because there is no time.

Point 2: My original comment: (i) “It is an attractive possibility that efficient thermal motion will realize a long step beyond several actin monomers along actin filament in a fine organized sarcomere of myofibrils” (line 116 to 119).

Author response: I described more details on lines 222-228. Under no load, thermal fluctuation of a weakly actin bound (A)M.ADP.P (non-force) may be uncoupled with ATP hydrolysis because the chemical reaction (ADP and Pi release) is slow as compared with shortening rate of myofibrils. The large fluctuation and actin-myosin dissociation-reassociation cycles of (A)M.ADP.P (nonforce) state may occur over several actin monomers. Only when forward stereospecific movement occurs, ADP and Pi releases occur from (A)M.ADP.P(force) to prevent backward movement.

My new comment: Please define “force” and “non-force” in parenthesis. Also, please explain how the speed of chemical reaction can be compared with the shortening rate of myofibrils. The unit of the former is /s, whereas the latter is μm/s, and therefore those two numbers cannot be directly compared. Please also define “stereospecific movement” and explain why stereospecific movement cannot occur in the backward direction in this model.

Response 2: The terms “force” and “non-force” are useful for isometric contraction, but confusing for free shortening under no load.  In non-force state, RLC domain thermally fluctuated and myosin heads detach and attach rapidly and move forward and backward at long distance.  When the myosin head moves forward at long distance, RLC domain is assumed to tilt forward to well-defined rigor-like (stereospecific) actomyosin-binding orientation (in force state) at a moment. At this moment, the phosphate release occurs and prevent backward movement by strong actin-myosin binding. When myosin head moves backward, the rigor-like binding orientation is assumed to be attained difficultly. This hypothesis was recently supported by single molecule observation of artificial thick filament (Fujita et al. communication biology, 2, 437, 2019)

    The ATPase measurement showed that the number of ATP hydrolyzed (/s) by one thick filament of one sarcomere (Z-line-digested myofibril) was much smaller than shortening rate (nm/s) divided by step size (<10 nm).

   I described them in lines 201-210.

Point 3: My original comment: (ii) “the actin may undergo conformational changes for contraction but not for regulation” (line 344 to 345).

Author response: I described details but weakly on lines 447-452. Our suggestion is that actin monomer does not undergo large conformational changes of the Ca2+-regulated thin filament. Ca2+-regulated movement of Tn-Tm complex may occur on a surface of actin monomer of the filament.

My new comment: I was unable to find the relevant information in lines 447-452, but am I right to assume that the author retracted the original provocative statement?

Response 3: Sorry for incorrect assignment.  Now you find it in lines 400-404.  

Point 4: My original comment: (v) Expressions such as (23/24 and 42/44) (line 199) needs explanations.

Author response: I put explanations for 23/24 and residues 23/27. They indicate the phosphorylation site and the distance between residues 23 and 27, respectively, on line 300.

My new comment: Something is wrong with the line numbers, and I cannot find the relevant explanations in line 300. Nonetheless, if 23/24 could mean either two residues or the distance between the two residues, that is rather confusing. In fact, “residues 23/27, located on the PKA phosphorylation sites Ser23/24, showed that the major distance distribution was markedly narrowed” (line 267) does not make sense to me.

Response 4: Sorry for incorrect assignment.  Now you find it in lines 271-273.  I also designated the phosphorylation sites as Ser23, Ser24 and Ser42, Ser44 and not use /.  The 23/27 was made clear by replacing it with the distance of paired residues 23/27. “The distance of paired residues 23/27, located on the PKA phosphorylation sites Ser23 and Ser24, showed that the major distribution was markedly narrowed” (lines 282-286).

Point 5: In addition, I found several minor problems in the newly added sections.

 (i) Please define the abbreviations in the first appearance. “CW” in line 40 is not defined. In contrary, some are doubly defined. Examples are FRET, SDSL and DQC.

Response 5: I fixed them on lines 40, 216-220, 348 according to the reviewer’s suggestion.

Point 6: (ii) Please explain “EPR suppress(es) the relative orientation selection for nitroxide probes” (line 68).

Response 6: I described more detailed explanation on lines 67-72. In FRET, the orientation between the donor and acceptor molecules must be known and this is difficult to know. EPR distance measurement does not have the problem about the relative transition dipoles because the localized magnetic transition dipoles are oriented by the magnetic field, not by the molecular structure. Furthermore, it is also possible to use non-selective pulses, minimizing effects of the relative label orientation, by detection of few traces at different magnetic fields.

Point 7: (iii) “This method is very useful for the arrangement of a-helices within protein domain” (line 72). “useful in elucidating the arrangement of…”?

Response 7: I fixed it on lines 75-76 according to the reviewer’s suggestion.

Point 8: (iv) Please explain “We believe that the observed changes both in the distance within 1.5 nm … arise from the backbone movement or rotation, and/or…” (line 92).

Response 8: I explained it on lines 95-101. When the conformational changes modulate tertiary interactions of sidechains that arise from the backbone movement or rotation, they should be detected by changes in spin label mobility: a striking pattern of changes in mobility and overall lineshape is observed. The distance change of nitroxide–nitroxide interactions (<1.5 nm) is ideal for mapping nearest neighbor secondary structures and their relative movement. The combined use of mobility and distance changes can provide a description of the conformational change at the level of the backbone movement and sidechain movement, although precise structural changes can not be determined.

Point 9: (v) “More recently, Arata et al. (unpublished [20]), using X-ray scattering in collaboration with the K. Wakabayashi group, proposed oppositely bent

structures as new intermediate states of ATP hydrolysis, similar to scallop ADP state [19]” (line 104-106). If the structure that Arata et al. proposed, based on a low resolution method, is similar to a previously resolved atomic structure of S1, how can they claim that it is a new intermediate? Speaking of an oppositely bent

structure, let me point out that Kimori et al. (2013) also reported such a structure (Biochem J, 450:23-35).

Response 9: I explained it in detail on lines 112-118 according to the reviewer’s comment. The structures of myosin head from SAXS during ATP hydrolysis at low temperature and the SH1-SH2 crosslinked state were similar to the crystal structure of scallop ADP state and electron microscopic structure of SH1-SH2 crosslinked state (Kimori et al. 2013) where the SH1-SH2 helix is unwound. The other previously solved intermediate state of crystal structures showed no indication of SH1-SH2 unwinding.  Because it is well known that during hydrolysis the SH1-SH2 helix is unwound, the oppositely bent structure must be a new structure for a part of intermediate states.

Point 10: (vi) “The central goal in muscle energy transduction has been to detect the force-generating structural changes in…” (line 136). “goal in muscle energy transduction research has been…”?

Response 10: I fixed it on line 148 according to the reviewer’s suggestion.

Point 11: (v) “Abe et al. [69] also determined the large-scale distance between the cTnC N-lobe and the C-lobe by DQC in the reconstituted cTnC-cTnI complex” (line 226-228). What is “large-scale distance”? 

Response 11: I deleted “large-scale” on line 244, according to the reviewer’s suggestion.

Reviewer 2 Report

The authors modified correctly in accordance with my comment about their unpublished results.  They used "the manuscript in preparation" or "conference proceeding" for "personal communication" except the #48 reference. Lines 705-711 and 751-755 in the authors' response to my comment 2) were not found in the revised manuscript. Lines 705-711 and 751-755 were in the references.

Author Response

Thanks for the helpful comments for my manuscript. Following the reviewer’s comments, I have addressed all the reviewer’s comments in order and, for each answer, I detail our changes made to the manuscript according to the reviewer’s comments.  I also highlighted them in the text of revised version. I hope that this will be acceptable for publication in the International Journal of Molecular Sciences.

Response to Reviewer 2 Comments

Point 1: Comments and Suggestions for Authors

The authors modified correctly in accordance with my comment about their unpublished results.  They used "the manuscript in preparation" or "conference proceeding" for "personal communication" except the #48 reference.

Response 1: I cited the #48 reference as personal communication, because this was originally cited as personal communication in the paper #45.

Point 2: Lines 705-711 and 751-755 in the authors' response to my comment 2) were not found in the revised manuscript. Lines 705-711 and 751-755 were in the references.

Response 2: Sorry for incorrect assignment.  Now you find it on lines 635-645 and 676-684. 

This manuscript is a resubmission of an earlier submission. The following is a list of the peer review reports and author responses from that submission.

Round 1

Reviewer 1 Report

This is a comprehensive review of very prolific career of an eminent EPR spectroscopist who pioneered EPR studies of motile systems.   Nevertheless, as a review of a topic it is seriously marred by: (a) uncritical acceptance of EPR line shape changes in terms of the backbone changes of a protein and (b) a one-sided bias towards author’s findings with no mention (bar three cases) of other researchers contributions, despite many competing studies in the field.

I will focus entirely on the first criticism since the second would be self-serving.  

All work that is reported here utilizes a single attachment point of a spin label to protein backbone.  Spin label is a analog of a sidechain albeit somewhat bigger.  The EPR signal arises from the very tip of the label, some 0.9 to 1.4 nm away from a backbone, linked to it by series of 4-5 single bonds. By definition not a rigid linkage and definitely more mobile than the backbone that except for the termini is immobilized by both ends.  Yet, in order to interpret spectral changes in terms of protein conformational changes one needs to assume that the linker is rigid – ie. we deny the sidechain a possibility of conformational change in order to invoke that change in the backbone.   That is simply not reasonable and there is a multitude of observations running against that assumption:  (i) in crystal structures, whenever the surface sidechains are resolved they have much higher B-factors than the backbone, most often electron density for distant atoms is not observed due to motion abolishing lattice sampling;  this is especially clear in crystal structures of spin labeled proteins;   (ii)  molecular modelling of spin labels reveals their mobility and multitude of conformations depending on the labeled sites,    (iii) perhaps most importantly the comparative studies using singly attached labels with the labels utilizing double attachment, or  metal ions rather than spin labels used here revealed 4-5 orders of magnitude higher mobility and broad distance distributions.   Sadly, there is no mention of those studies which bear direct relevance on the conclusions presented here and indeed were performed on identical systems.  Simply put, many of the observations of (a) distance changes smaller than 1.5nm or (b) multiple distance distribution, or (c) changing mobility that are the core of the observations in the review can be explained equally well by changes in spin label behavior and not the backbone.  

Reviewer 2 Report

The author has been studying conformational changes in various protein complexes using EPR and related methods, and in this review article, intends to summarize the results and present a principle that is common to those multiple protein machineries.  This is a very attractive goal, but I am afraid that a major revision is necessary before publication.  I would encourage the author to have one or two of his colleague to read the draft and incorporate the feedback for further improvements before re-submission.  Below is a list of my suggestions for the revision.

If this review is meant to be instructive for a broad range of readers interested in structure/function relationships of protein systems, the whole manuscript is too difficult to comprehend, in part due to poor organization.My research spatiality is with myosin molecular motors, and therefore, I was able to follow the parts on “2. Myosin ATPase” and “4. Kinesin ATPase”. However, this is because I had some background knowledge, and I could not understand other parts. For example, at the end of the “5. Clock ATPase” section, it was summarized as “The thermal conformational changes of KaiC tails precede the KaiA binding and auto-phosphorylation reaction” (line 509-511). However, I had no clue as to how this conclusion was reached based on the presented data.  The authors must provide sufficient background information, preferably with graphics, such that non-experts can understand the story.

Related to this, I found two very important and provocative statements regarding the acto-myosin system (listed below), but the important points are probably totally incomprehensible to non-expert readers.  If the author maintains the two points, he should explain the implications.

(i) “It is an attractive possibility that efficient thermal motion will realize a long step beyond several actin monomers along actin filament in a fine organized sarcomere of myofibrils” (line 116 to 119).

(ii) “the actin may undergo conformational changes for contraction but not for regulation” (line 344 to 345).

Apart from the organization problem mentioned above, many sentences are very difficult to understand, some of which have obvious grammatical problems. Below are just some of the examples:

(i) “A number of structural studies are focused on how myosin heads make a shape change on actin filament” (line 40-41).

(ii) “The spin label of intrinsic myosin heads showed considerable disorder associated with a well-defined orientation in muscle fibers in rigor state” (line 69-71).

(iii) “Takai et al. [99] measured the distance between spin labels near the stalk (Cys336) at NLs works as a strain gauge” (line 395-397).

(iv) “The transmembrane domain, which exhibits the Ca2+-binding movements of the three cytoplasmic domains with the conformation of this membrane domain induces the key events in this reaction cycle” (line 520-522).

(v) Expressions such as (23/24 and 42/44) (line 199) needs explanations.

I assume this manuscript is not meant to be an autobiography. If so, the author should pay more attention to be fair, and describe his own work in a broad context.Only with this, readers will be able to appreciate the scientific significance of the author’s group’s achievements, and to understand the strengths and weaknesses of the EPR-based methods. I noticed many frustrating statements, including:

(i) “Little is known on how the structural changes of myosin and actin are coupled to chemical reaction. The first suggestion was provided by Arata and Shimizu (1982) and later by Ostap et al.” (line 100-102). This is an absurd statement since many studies had been done to understand how ATP binding and hydrolysis are correlated with the structural changes of myosin before 1982.

(ii) Related to the docking/undocking of the neck-linker on to the kinesin motor core, the author develops arguments citing his own groups work as personal communication (Takai et al., 2012), but ignores similar EPR papers published nearly a decade before (e.g., Rice et al., Biophys J, 84:1844-54, 2003 and Sindelar et al., Nature Struct. Mol. Biol. 9:844-8, 2002). The author goes on to say “However, it is also suggested that NLs are two flexible to generate force, because undocked state exists at all the nucleotide states” (line 401-402).  Similar conclusion had been reached from thermodynamic considerations (Rice et al., 2003), which is not mentioned in this manuscript.

(iii) As the conclusion, the common theme is presented as “For energy transduction, thermal large-scale structural transition of this element from one to the other state precedes subsequent irreversible chemical reaction which stabilizes the latter state and prevents the reversal of the structural change” (line 608-611). This is a very important point, but the author is not the first to formulate this scenario. It is now established that the chemical state of an enzyme only modulates the equilibrium between multiple structural states, and that the product (e.g., ADP or Pi) release from an enzyme confers irreversibility to the cycle. The author should place the conclusion in those established contexts.

There are lengthy descriptions on the movements of RLC in muscle, and effects of phosphorylation on cTnC. However, they are not mentioned in the summary, and may be removed or condensed. In particular, the RLC movements section was boring and I was unable to get the take home message.

Many work from the authors group is cited as “personal communication”. Personal communication is usually used to describe unpublished information that the author obtained from some other researcher, and it is not a suitable method to present the author’s own unpublished data. It does not make any sense that the author is in the list of people who communicated with the author. Thus, I think all citation of such personal communications should be removed, and instead be shown as unpublished data. However, it should be noted that the mentioning to unpublished data should be kept to a minimum. I would encourage the author to publish such information as original research, if it is still worth it.If, on the other hand, the information had been reported by someone else, that report should be cited even if the publication was after the date the author’s group made the unpublished observation.

In addition to grammatical errors and typos, there are non-linguistic problems as well.

(i) “Arata et al. [5], using X-ray scattering in collaboration with the K. Wakabayashi group, proposed oppositely oriented bent structures as new intermediate states of ATP hydrolysis. Crystallographic studies finally showed them at atomic resolution [6]” (line 45-48). These statements imply that Arata et al. [5] discovered the new structure (but did not publish it) before the publication of the crystallographic study [6]. However, [5] is a personal communication in 2003, while the ref [6] was published in 2000 in PNAS. 

(ii) cTnC (line 157) -> TnC?

(iii) “black square” (line 327) -> “black circle”

(iv) “purple arrows” (line 433): where are they in the figure?

(v) In Fig 13 b, isn’t the position of the “WORK” wrong?  Shouldn’t it be associated with the leftward arrow at the bottom?

Figures reproduced from earlier papers should be stated as such.

Reviewer 3 Report

This review is an extremely wide-ranging overview on the mode of operation of a variety of ATPase systems, including motor protein (myosin), membrane transporting protein, and bacterial clock protein. It is especially valuable in stimulating interest of general readers to consider functional similarities in the above variable ATPase systems. Extensive references are also very valuable for those who have interest in energy transducing mechanisms. For the above reason, this review is worth publishing as part of IJMS special issue.

The author cites his own unpublished results as “Arata, personal communication” in the text and in References. It is inappropriate. The expression of “personal communication” should be changed to “unpublished results” in the text, and should be deleted from References.

The explanation of Fig.13, summarizing discussions, should be made much longer, so that this figure is made more comprehensive for general readers.